# Pharmacological Activities of Extracts and Compounds Isolated from Mediterranean Sponge Sources

**DOI:** 10.3390/ph14121329

**Published:** 2021-12-18

**Authors:** Lorenzo Di Cesare Mannelli, Fortunato Palma Esposito, Enrico Sangiovanni, Ester Pagano, Carmen Mannucci, Beatrice Polini, Carla Ghelardini, Mario Dell’Agli, Angelo Antonio Izzo, Gioacchino Calapai, Donatella de Pascale, Paola Nieri

**Affiliations:** 1Department of Neuroscience, Psychology, Drug Research and Child Health—Neurofarba—Section of Pharmacology and Toxicology, University of Florence, 50139 Florence, Italy; carla.ghelardini@unifi.it; 2Department of Marine Biotechnology, Stazione Zoologica Anton Dohrn, 80121 Naples, Italy; fortunato.palmaesposito@szn.it (F.P.E.); donatella.depascale@szn.it (D.d.P.); 3Department of Pharmacological and Biomolecular Sciences, University of Milan, 20133 Milan, Italy; enrico.sangiovanni@unimi.it (E.S.); mario.dellagli@unimi.it (M.D.); 4Department of Pharmacy, School of Medicine and Surgery, University of Naples Federico II, 80131 Naples, Italy; ester.pagano@unina.it (E.P.); aaizzo@unina.it (A.A.I.); 5Department of Biomedical and Dental Sciences and Morphological and Functional Imaging, University of Messina, 98125 Messina, Italy; carmen.mannucci@unime.it (C.M.); gioacchino.calapai@unime.it (G.C.); 6Department of Pharmacy, University of Pisa, 56126 Pisa, Italy; beatrice.polini@unipi.it (B.P.); paola.nieri@unipi.it (P.N.); 7Interdepartmental Center of Marine Pharmacology (MarinePHARMA), University of Pisa, 56126 Pisa, Italy

**Keywords:** Mediterranean sponges, marine natural products, antimicrobials, antiproliferative, anti-inflammatory, marine pharmacology, blue biotechnology, Mediterranean Sea

## Abstract

Marine pharmacology is an exciting and growing discipline that blends blue biotechnology and natural compound pharmacology together. Several sea-derived compounds that are approved on the pharmaceutical market were discovered in sponges, marine organisms that are particularly rich in bioactive metabolites. This paper was specifically aimed at reviewing the pharmacological activities of extracts or purified compounds from marine sponges that were collected in the Mediterranean Sea, one of the most biodiverse marine habitats, filling the gap in the literature about the research of natural products from this geographical area. Findings regarding different Mediterranean sponge species were individuated, reporting consistent evidence of efficacy mainly against cancer, infections, inflammatory, and neurological disorders. The sustainable exploitation of Mediterranean sponges as pharmaceutical sources is strongly encouraged to discover new compounds.

## 1. Introduction

Sponges represent a great reservoir of chemodiversity, playing a pivotal role in marine pharmacology [1], an exciting and growing discipline that blends blue biotechnology and the natural compound pharmacology. Although few representative compounds from sponges are approved as drugs, hundreds of new compounds with interesting pharmacological activities are currently in clinical trials and new molecules are discovered from sponges every year [2,3]. Sponge-derived products range from nucleosides to terpenes, sterols, cyclic peptides, and alkaloids and are proved to have pharmacological properties such as antibacterial, antiviral, antifungal, antimalarial, antitumor, immunosuppressive, and cardiovascular activity [1,4,5]. Their chemical diversity is the result of different factors. Sponges are invertebrates belonging to the Phylum Porifera, holding a unique internal system of afferent and efferent canals, functioning as a filter feeder [5]. By filtering both particulate and dissolved material, sponges fill a niche in nutrient cycling and play an important ecological role within the surrounded ecosystem [5,6]. They are amongst the most ancient animals in the oceans and can survive in a wide range of habitats. They developed, through years of evolution of adaptive strategies, to survive in different marine environments through the production of primary and secondary metabolites [5]. As sponges cannot move and lack of physical defenses, they have developed a broad suite of defensive chemicals to deter predators or compete for space [7]. Marine sponges also use their defensive chemicals to keep the offspring of tiny plants and animals (fouling organisms) from settling onto their outer surfaces [1]. It is important to highlight that marine sponges often host complex microbial communities composed of bacteria, archaea, microalgae, and fungi, representing 40% of the sponge volume. Sponges and symbionts evolution are closely linked [8] and this makes the real nature of the produced metabolites uncertain which could be attributed to the microbes or to the sponges, or even be the results of these strict interactions; beside this, sponges seem to be a perfect natural environment for the production of new metabolites. 

Several reviews report the state of the art natural products that have been discovered from marine sponges all over the world [1,5,9,10]. Nevertheless, an update on the natural products from Mediterranean sponges, which have been demonstrated to be a precious source of new molecules for pharmaceutical applications, is necessary. 

In this review, we summarize the up-to-date progress in preclinical pharmacology of the most promising Mediterranean sponge metabolites, by focusing on the most recent developments in the blue biotechnology field. We highlight the importance of the Mediterranean Sea for the exploitation of marine sponges, novel pharmacological advances, molecules identification, type of activity, mechanisms of action (when described), and the possible therapeutic uses of these naturally occurring molecules, focusing on antiproliferative, antimicrobial, anti-inflammatory, neurological, and other effects.

## 2. Methodology

Extensive bibliographic research was carried out by examining scientific articles that were published until 31 October 2021, in the major scientific databases and search engines of peer-reviewed literature on life sciences and biomedical topics (PubMed, Scopus, Embase, Web of Science, Google Scholar, The lens). The following combination of keywords have been applied for searching in the above-mentioned databases: “*Mediterranean Sponges*”, “*anti-inflammatory effects*”, “*cutaneous effects*”, “*neurological effects*”, “*metabolic effects*”, “*gastrointestinal effects*”, “*anticancer effects*”, and “*antimicrobial effects*”. The World Porifera Database (http://www.marinespecies.org/porifera/index.php, accessed on 16 September 2021) has been used as reference for statistical analyses setting the following parameters: *Geounit: Mediterranean Sea Area (General Sea Area); Distribution status to include: Valid; Synonyms: Only accepted names*. All of the published articles describing the biological effects of Mediterranean sponge-derived products were collected and discussed.

## 3. Mediterranean Sea Sponges

Sponges are characterized by a multitude of external shapes and colors and are mainly divided into four classes, namely Calcarea, Hexactinellida, Demospongiae, and Homoscleromorpha. This classification is based on their skeletal composition; Calcarea class, a species whose skeleton is formed by calcium carbonate (CaCO_3_); the Hexactinellida class, with a siliceous skeleton sponge (silicon dioxide SiO_2_); and the Demospongiae class (the most common, around 80–90%), with a skeleton that is mostly made up of a network of organic collagen from a structure-derived protein called spongin. Demospongiae include sponges of commercial interest for hygiene products and personal care and one of the most well-known is the Mediterranean sponge *Spongia officinalis*. Homoscleromorpha, the smallest class of Porifera, was originally considered part of Demospongiae, but further molecular studies allowed for a re-classification [11]. There are more than 9000 described and validated species that inhabit all the aquatic environments, although the actual number is expected to be more than double [12]. Table 1 shows the current numbers of accepted sponges that are recorded on the World Porifera Database (Accessed on 16 September 2021). Most of them are marine, around 98%, but also freshwater sponges are known. They are present at all water depths from the tidal zone to the deepest regions (abyss), as well as being widespread in the polar areas and tropical environments [13].

The Mediterranean basin has been ranked among the 25 “biodiversity hotspots” on earth on the basis of its high species richness and endemism [14,15]. Its semi-closed environment, the continuous enrichment of Atlantic waters (and species) and the Red Sea through the Suez Canal, and the mild climate makes it a perfect hotspot for biodiversity. Furthermore, its seabed, with numerous caves and deep canyons, constitutes an unexplored sponge biodiversity reservoir as demonstrated by Gerovasileiou et al., [16] or by the identification of carnivorous deep-sea sponges such as *Asbestopluma hypogea*, tiny sea sponges which have adapted to life in areas where food is scarce [17]. The predominant deep-sea group in the Mediterranean Sea is constituted by Hexactinellids, which are typically observed at bathyal and abyssal depths (i.e., below 200 m) [18].

Among dominant benthic taxa, Mediterranean sponge species account for over 600 with a high endemicity value (ca. 40%) [19,20,21,22]. As a matter of fact, an analysis through the World Porifera Database (http://www.marinespecies.org/porifera/porifera.php?p=checklist&gu_id=17805&inc_sub=1&rComp=%3E%3D&tRank=10&fossil=4&marine=1&status=cv&syns=vo&action=search (accessed on 16 September 2021)) resulted in 93 different families (Figure 1), which include more than 600 species and around 200 genera that are distributed in the whole Mediterranean Sea. Nevertheless, the real species richness of the Mediterranean Sea is still not clear or defined, encouraging future scientific exploration.

The high biodiversity rate of Mediterranean sponges at the taxonomic level is reflected in the wide range of bioactive extracts, fractions, and compounds that have been discovered and endowed with pharmacological potential. The use of sponges along the Mediterranean Basin dates from the Egyptian and Phoenician civilizations with evidence also in the Crete-Minoan culture [23] and Mediterranean sponges, that, even in present-day, have demonstrated to be a valuable resource of new pharmaceutically-relevant compounds. Figure 2 shows the different locations from where the reviewed sponges have been originally collected.

## 4. Pharmacological Actions and Potential Therapeutic Applications

Within the following sections, the activities and the molecules that have been discovered during the last 30 years have been reported, representing an update concerning the natural products that have been derived from Mediterranean sponges. Figure 3 highlights the main reported activities. As is clearly visible by the figure, anticancer and antimicrobial activity are the most studied/observed effects. In Table 2, the health benefits are summarized (with the sponges in alphabetical order) from all the results that were retrieved from the literature about the health beneficial properties of Mediterranean sponges. More than 50 natural products that have been isolated from the selected sponges are reported in Figure 4. In Table 2, the data are summarized as per sponge species, active compounds, effect, and efficacious concentration/dose.

### 4.1. Anticancer Effects

Marine sponges produce themselves, or in association with the guest symbionts, cytotoxic, mainly proapoptotic, protective molecules against pathogens, predators, and parasites [65]. Some of these molecules have been demonstrated to be powerful tools against cancer. For example molecules that were extracted from *Demospongiae*, called spongosines, i.e., spongouridine and spongothymidine, have been used as lead compounds for the synthesis of cancer chemotherapeutic agents [66,67], cytarabine (Cytosar-U^®^), fludarabine phosphate (Fludara^®^), and nelarabine (Atriance^®^). These agents are analogues of spongosines that are marketed as anticancer agents which act as antimetabolites, blocking nucleic acid polymerization [66,67]. In an advanced clinical trial, plocabulin was also investigated, a polyketide that was isolated from the Madagascan sponge *Lithoplocamia lithistoides* [67]. Regarding sponges from the Mediterranean area, the first pharmacological study investigating the potential anticancer activity was published in 1985 [44]. Müller and collaborators isolated two secondary sesquiterpenoid metabolites that are avarol (**1**) and its derivative avarone (**2**) from *Dysidea avara* that were collected in the bay of Kotor, ex Jugoslavia [68]. They reported a cytostatic activity in L-5178y leukemia cells and a good antileukemic effect in vivo in mice where an i.p. administration of 10 mg/kg, for five days, showed avarone to be more efficacious than avarol (70% and 20% inhibition of tumour growth, respectively). In subsequent years, avarol and avarone cytotoxic effects along with (−)-3′-methylaminoavarone (**3**), (−)-4′-methylaminoavarone (**4**), and N-methylmelemeleone-A (**5**), showed cytotoxic effects with IC_50_ values in the low micromolar range [43,45,46]. A molecular mechanism was suggested for avarol-induced apoptosis in pancreatic ductal adenocarcinomas (PDACs) [69], where it was observed an endoplasmic reticulum (ER) stress-dependent apoptosis through the PERK–eIF2α–CHOP signaling pathway. In addition, Ciftci et al. [42] reported the pro-apoptotic activity of an ethanolic extract from *D. avara* that were collected from Dardanelles (Turkey), that appeared to be associated with the inhibitory ability against a panel of tyrosine kinases, particularly the intrinsic kinase of platelet-derived growth factor receptor (PDGFR)-b. 

Other cytotoxic sesquiterpenoids, called paniceins, were obtained from *Reniera fulva* sponge that were collected near Favignana island (Italy) [57]. Among them, paniceins B3 (**6**) and C (**7**) inhibited cell viability of non-small lung cancer (NSLC) and leukemia cells, while panicein A (**8**) hydroquinone revealed selective cytotoxicity against leukemia cells. Paniceins were first described 20 years earlier, extracted from the *Halichondria panicea* that were collected in the bay of Naples (Italy) [70], but they were not evaluated from a pharmacological point of view. 

Another Mediterranean sponge containing panicein hydroquinones is *Haliclona* (*Soestella*) *mucosa*, collected in a cave at Villefranche-sur-Mer (France) [51]. The authors reported that panicein A hydroquinone (**9**) had a low cytotoxic effect, per se, but inhibited the Patched efflux activity and enhanced doxorubicin cytotoxicity on melanoma cells.

In addition, polyprenyl-hydroquinones and furanoterpenoids were described from *Spongia officinalis*, *Ircinia spinulosa*, and *Ircinia muscarum*, that were derived from different locations in the Mediterranean sea (in Bodrum, Turkey; in the bay of Naples, Italy; at Zadar, Croatia; and, only for *Ircinia muscarum* at Baleares, Spain) [54]. These compounds, 4-OH-3 tetraprenylphenylacetic acid (**10**) and 2-Octaprenylhydroquinone (**11**), showed low micromolar potency as inhibitors of the cell division cycle 25 (CDC25) phosphatase [54], a protein family which represents a relevant target for cancer therapy [71]. Previously, the furanoterpenes from *Spongia officinalis,* furospongin 5 (**12**), demethylfurospongin 4 (**13**), and cyclofurospongin 2 (**14**) that were collected from La Caleta (Cadiz, Spain) were shown to exert only a weak cytotoxic effect in one (i.e., the P-388 mouse lymphoma cell line) out of the four cell lines that were tested [63]. Antiproliferative activity against three human cancer cell lines by two fractions of methanolic extracts from *Spongia officinalis* that were collected in various areas of the coastal region of Monastir (Tunisia) was also reported, but any identification of the pure active compounds was done [62]. 

Polyprenyl-hydroquinones, hexa- (**15**), hepta- (**16**), and nona- prenyl-1,4-hydroquinone (**17**), purified from different Mediterranean sponges of the genera *Ircinia* and *Sarcotragus* i.e., *S. spinosulus* that were collected from Callejones (Spain); *S. muscarum* from Mersin (Turkey); and *Ircinia fasciculata* from Fethiye (Turkey), were investigated for their potential anticancer activity in other works [53,61]. In *S. spinosulus*, a new hydroxylated nonaprenyl-hydroquinone (**18**), along with the known metabolites hepta- and octa-prenylhydroquinones, were evaluated on a chronic myelogenous leukemia (CML) cell line and the latter two exhibited a potency ≤ 10 µM [61]. From *S. muscarum* and *Ircinia fasciculata*, three polyprenyl-1,4-hydroquinone derivatives were purified [53]. Among them, the heptaprenyl-hydroquinone was the most cytotoxic on hepatoma cells, revealing a proapoptotic mechanism with the ability to inhibit NF-kB signaling and several protein kinases involved in cancer progression [53].

In another work [60], three prenylated-hydroquinones (**15**, **16**, **17**), in addition to three known furanosesterpene tetronic acids (iricinin 1 (**19**), sarcotin A (**20**), and variabilin (**21**)) were purified from three *Sarcotragus* sponge species that were collected from Tunisian coasts: *S. spinosulus* from Tabarka and Monastir; *S. fasciculatus*, collected from Monastir; and *S. foetidus* collected from Cap Zebib. Comparing the inhibitory activity of the six compounds on viability of a panel of cancer cells with different sensitivity to pro-apoptotic stimuli, the polyprenyl-hydroquinones displayed greater potency than furanosesterpene tetronic acids, and only the hydroxylated polyprenyl-hydroquinone showed IC_50_ < 10 M in all cell lines that were tested [60].

Another interesting class of natural molecules that have been extracted from sponges are alkaloids. In sponges of the Mediterranean area, two families of marine alkaloids, specifically guanidine alkaloids, named crambescidins and crambescins, were described in the early 1990s from the same research group. The authors reported an antileukemic effect of crambescidins (800 (**22**), 816 (**23**), and 844 (**24**)) that were extracted from the sponge *Crambe crambe* that were collected at Isla de Formentor (Cueva, Spain) [38,72]. More recently, crambescidin proapoptotic effects were also reported from a human colon cancer-xenografted zebrafish model by Roel et al. [39]. Interestingly, non-cytotoxic concentrations of crambescidins 816 (**23**) (1 and 2 µM) significantly decreased tumor growth and increased the survival rate of zebrafish embryos (transplanted with human colorectal carcinoma cells). In vitro experiments aiming at investigating the mechanism of action found that three crambescidins (830 (**25**), 800 (**22**), and 816 (**23**)) strongly down-regulated cyclin-dependent kinases 2/6 and cyclins D/A expression, inhibited tumor cell adhesion, and altered cytoskeletal integrity, promoting the activation of the intrinsic apoptotic pathway [39]. 

Another member of the Crambeidea sponge family, *Crambe tailliezi,* collected in the Mediterranean Sea (unspecified area), after a dichlorometane-methanolic extraction, revealed a high molecular weight molecule that was named the P3 compound (**26**). This compound inhibited the mitotic kinases Aurora A and B and induced apoptotic cell death in human osteosarcoma cells but did not appear to be cancer cell-selective [40]. A cytotoxic alkaloid that was identified as the 4S-guanidino-pyridazine compound, named zarzissine (**27**), was obtained instead from a dichloromethane extract of the sponge *Anchinoe paupertas* that was collected at Zarzis (Tunisia) [27]. Several alkaloids were also obtained from the chemical investigation of the diethyl ether extract from the marine sponge *Rhaphisia lacazei* that was collected in the “Grotta dei Gamberi” (Ustica, Italy) [56]. Out of 13 of the isolated bisindole alkaloids, two compounds belonging to the class of topsentins, topsentin B1 (**28**) and B2 (**29**), showed cytotoxic activity in vitro against human broncopulmonary cancer cells [56]. 

From the sponge *Agelas oroides* that was collected in the Maltese Sea (at Rdum il Bies) Konig et al. identified five bromo-pyrrole alkaloids, among which oroidin (**30**) and the most active against cancer cells was the 2-cyano-4,5-dibromopyrrole compound (**31**) [25]. More than 15 years later, Dyson et al. starting from the bromo-pyrrole alkaloid oroidin, developed another 30 compounds and three of them showed cancer cell cytotoxicity with IC_50_ less than 5 µM [73]. 

Isofistularin-3 (**32**), another bromo-alkaloid (bromotyrosine) from *Aplysina aerophoba* collected in the Mediterranean Sea (unspecified area), was first investigated as a potential anticancer agent by Florean et al. [29]. Similar to other sponge-derived bromotyrosines that were previously observed to inhibit methyltransferase and histone deacetylase enzymes [74,75,76,77], isofistularin-3 inhibited DNA methyltransferase (DNMT)1. In silico research demonstrated its binding with the DNA interacting pocket of the enzyme. Moreover, it induced autophagy and it was also a sensitizing agent in combination treatments with a tumor-necrosis-factor-related apoptosis inducing ligand (TRAIL). In addition, many other intracellular changes (i.e., G0/G1 cell cycle arrest, increased p21 and p27 expression, as well as reduced cyclin E1, PCNA, and c-myc levels) that were induced by the compound were observed and no toxicity on peripheral blood mononuclear cells (PBMCs) from healthy donors and on zebrafish development was observed [29]. 

Recently, isofistularin-3, that was extracted from *Aplysina aerophoba* collected in an aquaculture facility in the Adriatic Sea (Kotor Bay, Montenegro), was evaluated for its anti-tumoral activity by Binnewerg et al. [28]. After purification from a methanolic extract, isofistularin-3 and aeroplysinin-1 (**33**), another bromotyrosine, inhibited cell viability in a cell-dependent manner. Specifically, aeroplysinin-1 inhibited the growth of a neuroblastoma cell line and isofistularin-3 inhibited the growth of a breast cancer cell line. Moreover, in contrast with aeroplysinin-1, isofistularin-3 did not affect the growth of healthy cells (mouse endothelial cells and fibroblasts). 

In addition to the terpene-like compounds and alkaloids that have been previously described, several other miscellaneous compounds that are produced by Mediterranean sponges were identified and studied for their potential anticancer activity. 

In the paper of Ferretti et al. [24], the methanolic extracts from *Petrosia ficiformis* and *Agelas oroides,* both from Portofino’s promontory (Italy) were investigated. While the *Petrosia ficiformis* extract induced a sustained cytotoxic effect at any concentration or time exposure in neuroblastoma cells, the *Agelas oroides* extract in the same cells provoked an apoptotic effect and a reactive oxygen species production (ROS) in a concentration-dependent manner.

In a methanolic extract from *Geodia cydonium*, collected in the Gulf of Naples (Italy), an active fraction that was able to induce cellular apoptosis while remaining inactive on normal breast cells was identified by Costantini et al. [49]. Changes by the fraction in amino acid metabolism, as well as modulation of glycolysis and glycosphingolipid metabolic pathways were observed without identifying the chemical component(s) that was responsible of the activity [49]. 

In a recent paper, from the Demospongia *Chondrosia reniformis* that were collected in the area of the Portofino Promontory (Liguria, Italy)*,* a new protein named chondrosin (**34**) was identified starting from a crude hydrophilic extract [35]. Chondrosin showed anti-tumoral activity and more significant toxicity against murine leukemia cells than against human cancer cells. Extracellular calcium intake followed by an increase in cytoplasmic ROS was the cell death mechanism that was recognized [35].

### 4.2. Antimicrobial Effects

Antimicrobial resistance (AMR) is one of the biggest threats to human and animal health today [78]. It has reached a crisis point with both common and life-threatening infections becoming increasingly untreatable. To counteract this phenomenon, the hunt for new antibiotics is spearheaded by natural products (NPs) discovery. Sponges, which are the most primitive invertebrates, are considered an interesting source for the discovery of antimicrobial substances. In fact, these animals are frequently exposed to intense predation by selected groups of marine animals such as turtles, sea urchins, and sea stars, as well as tissue infection by microorganisms. Sponge secondary metabolites clearly present a defensive role against predation [79].

Numerous sponges that have been collected from the Mediterranean Sea during the last decades have been demonstrated to exert antimicrobial activities against a wide range of human pathogen microorganisms, especially bacteria and fungi. The most common approach that is applied to evaluate the sponge’s antimicrobial activity is bioassay-guided fractionation which has often led to the isolation and chemical identification of the active molecules. Extensive research has been carried out to unveil the sponge-derived antimicrobial compounds, including different genera such as *Axinella, Ircinia, Agelas,* and *Dysidea*, with results believed to be promising [80]. Marine sponges of the genus *Axinella* contains approximately 20 species that are distributed worldwide and it is known to be a source of a variety of secondary metabolites, such as brominated pyrrole alkaloids, cyclopeptides, polyethers, sterols, and terpenes [81,82,83]. 

Yassin et al., [33] identified antimicrobial compounds from *Axinella verrucosa* (a Demospongia that is present in the whole Mediterranean) collected in the Latakia coast (Syria). In their experiment, dry sponge pieces that were previously cleaned by water to remove epibionts, were processed by sequential extraction, decreasing the solvent polarity. The Kirby-Bauer Antimicrobial Assay showed a potent inhibition effect (>20 mm of inhibition zone), comparable to common antibiotics, of the methanolic extract against clinical specimens *Staphylococcus aureus*, *Pseudomonas aeruginosa*, *Acinetobacter septicus*, and *Proteus vulgaris*. By mass spectrometry LC-ESI-MS and UV spectroscopy, three pyrrole imidazole alkaloids, namely hymenialdisine (**35**), and two derivatives, 10-E-hymenialdisine (**36**) and spongiacidine B (**37**) were identified. The antimicrobial effect of the *A. verrucose* extract was attributed to the synergistic actions of the three molecules. Consistently, a previous investigation showed that the ethyl acetate extract of *A. damicornis,* collected from coastal water of Monastir (Tunisia), inhibited the growth of *P. aeruginosa* and the gentamycin-resistant strains *Listeria monocytogenes* and *Enterococcus feacalis* as well as other human pathogen bacteria, with an inhibition zone from 7 to 26 mm [26]. In the same study, another Demospongiae, *Agelas oroides,* collected in the same environment, showed a broad range of antibacterial activity and moderate activity against fungal strains. The antimicrobial activity of sponge *A. oroides* extracts was performed using the agar-disk diffusion assay. Both *Axinella* and *Agelas* extracts contain a complex mixture of structurally different brominated pyrrole alkaloids. Sponges of the family *Agelasidae* produce oroidin (**30**) alkaloids which mainly serve as a chemical anti-feeding defense mechanism against predators. This compound interferes with the biofouling process of the bacterium *Rheinheimera salexigens*, retarding the bacterial attachment and colonization [84]. Recently, a metabolomics approach by mass spectrometry allowed the evaluation of the Mediterranean marine sponge *A. oroides* chemical diversity with antibiofilm activity. The authors identified 13 known oroidin class alkaloids along with one new monobromoagelaspongin, five betaines and one amine. One of those compounds, (−)-equinobetaine B, was found to be an enantiomer of the known natural product (+)-equinobetaine B [85].

In addition to *Axinella* and *Agelas*, a relevant source of brominated biologically- active compounds is the genus *Aplysina*, formerly known as *Verongia*, one of the richest genera in terms of secondary metabolites [86], with *A. aerophoba* and *A. cavernicola* present in the Mediterranean Sea [86]. Among all discovered compounds, (+)-aeroplysinin-1 (**33**), isolated for the first time in *Verongia* and then found in other *Aplysina* sponges (included the Mediterranean ones), is a chemical weapon to protect sponges from pathogens and predators. It has potent antibiotic effects on Gram-positive bacteria and several dinoflagellate microalgae causing toxic blooms. In preclinical studies, (+)-aeroplysinin-1 has been shown to have promising anti-inflammatory, anti-angiogenic, and anti-tumor effects. Due to its broad pharmacological spectrum, (+)-aeroplysinin-1 is believed to have a pharmaceutical interest for the treatment of different pathologies [30]. 

Another sponge that is distributed in the Mediterranean Sea is *Haliclona fulva,* a red Demospongia, otherwise known as *Fulva aliclona*. The butanolic extract of this species, collected at 40 m depth in the area of Procida Island (Gulf of Naples), contained nine linear polyoxygenated acetylenes, fulvynes A-I (**38–46**) [50]. The new compounds are characterized by a long linear alkyl chain bearing a residue of propargylic acid, a terminal acetylenic moiety, a diacetylenic carbinol, and several hydroxyl and keto groups. In-depth structural elucidation was performed by using HRMS, NMR, and other spectroscopic techniques. All fulvynes were found to be active against a *Bacillus subtilis* chloramphenicol-resistant, with IC_50_ values between 60–12 µM. Not so far from Procida island, in the bay of Naples, the Demospongia *Dysidea avara* was collected [68] from which the potent, avarol (**1**) (a novel sesquiterpenoid hydroquinone with a rearranged drimane skeleton) and its derivative avarone (**2**) were isolated. Avarol possesses a rigid sesquiterpene skeleton and a reactive hydroquinone moiety, which can interfere with reactive oxygen species production and the redox status of cells and exerts antimicrobial, cytostatic, anti-HIV virus, anti-inflammatory, and antiparasitic activity [87]. More recently, Pejin et al., [47] extended the in vitro screening of antimicrobial activity of avarol, demonstrating its anticandidial activity by microdilution method against eight *Candida* strains; this compound was proven to be active against all the strains that were tested (ranging from MIC 0.8–6.0 μg/mL and MFC 1.6–12.0 μg/mL, respectively). Compared with the standard antifungal drug fluconazole (MIC 0.5–30.0 μg/mL and MFC 1.0–60.0 μg/mL), avarol possesses similar or higher anticandidal activity. 

In addition to Dermospongiae, the class of Calcareous sponges are known to produce a wide range of bioactive compounds. Among them, *Clathrina clathrus*, a Calcispongia collected in the northwestern Mediterranean Sea (Marseille, France), produces antibacterial and antifungal compounds [36]. CH_2_Cl_2_/MeOH extract of *C. clathrus* followed by purification and 1D and 2D NMR spectroscopy revealed the presence of a new 2-aminoimidazole alkaloid, named clathridimine (**47**), along with the known clathridine (**48**) and two minor metabolites preclathridine (**49**) and clathridine-zinc complex. Clathridimine displayed selective anti-*Escherichia coli* (15 mm inhibition zone) and anti-*Candida albicans* (24 mm inhibition zone) activities, evaluated by the agar diffusion assay. Clathridine showed anti-*Candida albicans* activity only and its zinc complex exhibited selective anti-*Staphylococcus aureus* activity. A large diversity of 2-aminoimidazolone alkaloids is produced by various marine invertebrates, especially by the marine Calcareous sponges *Leucetta* and *Clathrina* [88]. 2-aminoimidazolones have been proven to possess antibacterial activity, anti-biofilm, antifungal, and many other bioactivities [88]. Analogs of the 2-amino-imidazole derivatives were isolated from other calcareous sponges belonging to the genus *Leucetta* collected from the Pacific Ocean and the Red Sea, raising the question of their biosynthetic origin if they were directly produced by the sponge or by associated microorganisms; however, studies by Roue et al., (2010) suggest that the real producer is the sponge. In fact, in their work the authors performed a sponge dissociation and cell fractionation, separating the sponge cells from the bacteria by differential centrifugation followed by trypsinization of the sponge cell surface and detected, by chemical analysis with HPLC/UV/ELSD, that clathridine was localized in the sponge cells [36].

Proof regarding the enormous potential of Mediterranean sponges as a source of new antimicrobial compounds has been recently highlighted by papers that focused on the Turkish marine environment [80,89,90]. Within this context, Konuklugil and Gözcelioğlu [80] evaluated the antimicrobial activity of 33 methanolic extracts (and their fractions) that were derived from a large group of sponges that were collected in different locations in the Turkey sea, most of which are widely distributed in the whole Mediterranean. The extracts and fractions (250 µg/mL) were screened towards two Gram-positive bacteria methicillin-resistant, *S. aureus* (MRSA) and vancomycin-resistant *Enterococcus* (VRE); one yeast *C. albicans*; and two Gram-negative bacteria, *P. aeruginosa* and *Proteus vulgaris.* Strong to modest inhibitory effects were observed depending on the area of the sponge collection, however the active compounds were not identified. This suggests that the production of secondary metabolites could also be affected by the surrounding environment and ecosystem. A recent review of the literature highlighted relevant findings of novel bioactive compounds from sponges that were collected close to Turkish coasts [90], indicating *Dysidea* sp., *Agelas* sp., *Spongia* sp., *Ircinia* sp., *Sarcotragus* sp., and *Axinella* sp., as the most promising sources of secondary metabolites. Methanolic extracts of these samples showed an inhibition effect. In particular, *Aplysina aerophoba* and *Spongia agaricina* significantly inhibited a wide range of bacteria and fungi.

### 4.3. Anti-Inflammatory Effects

Marine sponges produce a wide range of metabolites that have been identified as anti-inflammatory agents. The anti-inflammatory activity was firstly disclosed for avarol (**1**) and avarone (**2**) from *Dysidea avara*. Avarol and avarone potently reduced mouse paw edema after both oral (ED_50_: 9.2 and 4.6 mg/kg, respectively) and topical (97 and 397 mg/ear, respectively) administration [41]. Additionally, avarol and avarone inhibited leukotriene B_4_ (LTB_4_) and thromboxane B_2_ (TXB_2_) release as well as superoxide production in A23187-stimulated rat peritoneal leukocytes, with an IC_50_ value below the mM range. Among these two marine metabolites, avarol was also able to inhibit human recombinant synovial phospholipase A2 (PLA2) activity (IC_50_ = 158 µM) in stimulated leukocytes [41].

PLA2 hydrolyzes the fatty acids from the *sn*-2 position of membrane phospholipids, being responsible for arachidonic acid release and the consequent biosynthesis of eicosanoids and related bioactive lipid mediators [91]. In addition to avarol and avarone, other marine Mediterranean sponge-derived compounds have been shown to affect PLA2. For example, two 2-polyprenyl-1,4-hydroquinone derivatives, namely 2-octaprenyl-1,4-hydroquinone (**11**) (IS2) and 2-[24-hydroxy]-octaprenyl-1,4-hydroquinone (**50**) (IS3), isolated from the marine sponge *lrcinia spinosula* that was collected in the bay of Naples (Italy), inhibited human recombinant synovial PLA2 in a concentration-dependent manner, an effect that was associated to a reduction of LTB4 and TXB2 biosynthesis and release in human stimulated neutrophils (IC_50_ values in the micromolar range) [55]. Moreover, topical administration of IS2 and IS3 reduced the murine ear experimental inflammation, with IS3 more active than IS2 in inhibiting leukocyte migration as revealed by the myeloperoxidase activity assay. In a further study, carvenolide (**51**), a C_21_ terpene that was isolated from *Fasciospongia cavernosa* (collected in the Aegean Sea), inhibited in vitro human synovial PLA2 in a concentration-dependent manner (IC_50_: 8.8 µM) and reduced TNFα, nitrite and prostaglandin E_2_ (PGE2) production in zymosan-stimulated raw 264.7 macrophages [48]. Such effects were associated with dual inhibition of inducible nitric oxide synthase (iNOS) and cyclooxygenase (COX)-2 expression [48].

### 4.4. Neurological Effects

Neurological diseases are a heterogeneous group of disorders of the peripheral and central nervous system which include Alzheimer’s and Parkinson’s diseases, multiple and amyotrophic lateral sclerosis, neuropathies, epilepsy, stroke, psychiatric disorders, and many others. As a common feature, multiple biochemical pathways are dysregulated, which complicate the medical approach and make the current pharmacotherapy unsatisfactory [92]. Thus, naturally occurring sources, including Mediterranean sponges, might offer new ideas for pharmaceutical development.

*Spongia officinalis* (collected in the coastal region of Tunisia) contained crude and methanol-water extracts that showed anticonvulsant and analgesic effects in mice [64]. Specifically, the filtered and lyophilized raw material was compared with four fractions that were obtained by methanol-water extraction with an increasing quantity of methanol (0, 25, 50, and 80%). The crude extract (600 mg/kg, s.c.) and 50% methanol fraction (100 and 200 mg/kg s.c.) delayed the onset and decreased the duration of pentylenetetrazole-induced seizures after a single administration. Further, all *Spongia* fractions (with higher efficacy and potency showed by the 50% methanol fraction) reduced irritative pain that was induced by acetic acid or phenylbenzoquinone administration (writhing test). Unfortunately, no data about the active ingredients that were responsible for such activities were provided.

Neuroprotective in vitro activities by components of *Axinella verrucosa* (collected in the Corsica island) were described [32]. Methanol/chloroform extract was fractionated until the purification of bromopyrrole alkaloids. In cultures of primary neurons that were obtained from the rat cortex, *A. verrucosa* alkaloids (among others compounds 1–4, 10 µg/mL) reduced the intracellular increase of free Ca^2+^ evoked by serotonin, glutamate, or quisqualic acid, thus limiting neuronal excitotoxicity. The same authors isolated two novel bromopyrrole alkaloids, damipipecolin (**52**) and damituricin (**53**) from *Axinella damicornis* (collected in the Corsica island) [31]. Both of the compounds counteracted serotonin-induced intracellular Ca^2+^ enhancement, with a distinctive shape bell curve peaking at 0.1 µg/mL.

In 1997, Sepčić et al. isolated 3-alkylpyridinium oligomers and polymers (**54**) from *Reniera sarai* (collected in the northern Adriatic Sea). The aqueous sponge extract contained 3-alkylpyridinium polymers (poly-APS) ranging from 5520–18,900 Da. Poly-APS inhibited the activity of acetylcholine esterase (AChE) that was obtained from different sources. Specifically, human erythrocyte AChE, electric eel AChE, insect recombinant AChE, and horse serum butyrylcholinesterase were inhibited by 50% from 0.06, 0.08, 0.57, 0.14 µg/mL poly-APS, respectively [58]. More recently, poly-APS was shown to inhibit nerve-evoked isometric mouse skeletal muscle twitch and tetanic contractions (IC_50_ = 29.4 μM and 18.5 μM, respectively) and to produce a 30–44% decrease of directly muscle-elicited twitch and tetanus amplitudes at 54.4 μM [59]. Additionally, poly-APS (9.1–27.2 μM) markedly decreased the amplitude of miniature endplate potentials, with their frequency to be affected at the highest concentration used; a block of neuromuscular transmission by a non-depolarizing mechanism was suggested. To note, cardiovascular side effects were reported in rat ex vivo evaluation using poly-APS higher than 10 nM [93].

### 4.5. Other Effects: Cutaneous, Metabolic, and Gastrointestinal

*Cutaneous.* The skin is the largest organ of the human body and provides defense against harmful external factors such as mechanical and chemical insults, heat, infections, water loss, and ultraviolet radiation. To date, only one work has investigated the effects of Mediterranean sponge products in skin diseases.

HPLC-purified fractions of trypsin-digested collagen extracts from the marine sponge *Chondrosia reniformis* (MCH, marine collagen hydrolysates), collected from the area of the Portofino promontory (Liguria, Italy), were tested on keratinocytes, the most abundant cells of the epidermis (the external layer of the skin, which also contains other cell types, such as melanocytes, Langerhans, and Merkel cells) and on fibroblasts, cells that are involved in matrix formation and in tissue repair (typically found in dermis). 

Four MCH fractions differentially enhanced the proliferation of L929 fibroblasts and HaCaT keratinocytes (between 1.29- and 1.63-fold increase, starting from 10 µg/mL), showing ROS scavenging activities in DPPH and NBT/riboflavin test, too. All MCH fractions enhanced fibroblast collagen 1A (expression and release at 100 µg/mL) and protected keratinocytes and fibroblasts from UV exposure (50 µg/mL). Finally, all the treatments reduced keratin 1 and 10 expression in keratinocytes, two markers of skin thickening and reduced elasticity, especially those fractions containing peptides with higher concentration of hydroxylproline. In both cell types, the hydroxylproline-rich fractions demonstrated promising wound-healing properties, facilitating either cell migration or proliferation [34]. 

*Metabolic.* Several sponge preparations and isolated compounds exert pharmacological actions, which might be predictive of antidiabetic actions. These include inhibition of glycogen synthase kinase 3β (GSK-3ß), alpha-glucosidase, PTP1B, dipeptidyl peptidase IV, or protection of the beta pancreatic cells. 

The enzyme GSK-3β plays a relevant role in the pathogenesis of diabetes. The development of selective inhibitors of this enzyme is believed to be an attractive strategy for the development of potential new antidiabetic drugs. A sesquiterpene compound named palinurin (**55**), deriving from the sponge *Ircinia dendroides,* exerted GSK-3 ß inhibitory activity (IC_50_ 2.6 µM) [52]. Furthermore, GSK-3ß inhibitors were also isolated from the marine sponges *Ircinia variabilis* and *Ircinia oros.*

*Gastrointestinal.* The gastrointestinal tract represents an important barrier between human hosts and microbial populations. One potential consequence of host-microbial interactions is the development of mucosal inflammation which can lead to gastrointestinal ailments including gastritis, ulcer, and inflammatory bowel diseases. Gastrointestinal diseases are emerging pathological conditions whose prevalence has increased in the last few years; thus, the search for new active compounds to counteract these inflammatory conditions, also from natural sources, is advisable. Berlinck et al. evaluated the effects of pure compounds from marine sponges in the gastrointestinal tract. A total of four guanidine derivatives were isolated from the marine sponge *Crambe crambe,* which is widely present along the rocky coasts of the Mediterranean area (Favignana island). Among these, crambescidin 816 **(23)** inhibited acetylcholine-induced contraction of guinea pig ileum at very low concentrations (−30% at 6 pM). Accordingly, this compound was a potent antagonist of voltage-sensitive Ca^++^ channels (1.5 × 10^−4^ µM) neuronal cell line (NG-108-15) [37]. 

## 5. Conclusions and Perspectives

Sponges are among the most prolific producers of a huge number of bioactive compounds and have the potential to provide future drugs against severe diseases. In fact, the biological and therapeutic activities of new metabolites from sponges have been reported for years in scientific publications. Although they are the oldest, still living, multicellular animals on this planet and widespread within aquatic ecosystems, they have been poorly studied in comparison with the recognized ecosystems such as corals, algae, and fish. This is especially true for Mediterranean sponges, even if it represents one of the most biodiverse marine habitats. Within the present review, a significant body of evidence regarding the preclinical research of sponge-derived products with anticancer and antimicrobial activities has been reported. To summarize, more than 20 different species of sponges that were collected in the Mediterranean Sea were revealed to contain metabolites with antiproliferative/cytotoxic activity on cancer cells. Recognized compounds that mainly belong to terpene-like and alkaloid classes may be the active compounds, while apoptosis is the main mechanism of cell death that was observed. The intracellular mechanisms were investigated only in a few papers, and, in many cases, protein kinase-inhibition was suggested to be involved. Concerning the antimicrobial activity, the promising studies, especially from the Turkish coasts, demonstrated the antimicrobial effects of a high number of sponge crude extracts or fractions, encouraging the research of active metabolites also in other Mediterranean areas and indicating the need to further purify and characterize them. The literature analysis indicates that the drug discovery, especially from *Axinella*, *Agelas*, *Haliclona*, *Dysidea*, *Aplysina,* and *Spongia*, together with many other genera and species, is far from being concluded. Anti-inflammatory and neuromodulatory effects were also reported but although they show a relatively good potency against specific targets, the number of studies is still limited. Furthermore, sporadic, albeit interesting studies were dedicated to cutaneous, metabolic, and gastrointestinal applications. Among all of the reported compounds, avarol and its derivatives displayed a wide range of bioactivities, as well as aeroplysinin-1, and extracts containing alkaloids. 

Besides this clear biosynthetic potential, one of the major issues which hampers the economic exploitation of sponge-derived products is related to the limited availability of larger quantities of sponge material, also defined as “the supply problem” [94,95]. This problem highlights the small number of chemical metabolites that are present within the sponge compared to their biomass, which, combined with other factors such as the dilution effect of the sea water on target compounds, makes the industrial exploitation of sponges very challenging. In the era of the Ocean Decade (2021–2030) [96], which aims to ensure that ocean science delivers greater benefits for both the ocean ecosystem and for society, harvesting large amount of marine biomass is not acceptable and not sustainable. Therefore, new and innovative cultivation approaches need to be developed for the sustainable exploitation of marine sponges as pharmaceutical sources. For this purpose, different approaches have been applied so far which mainly include the in situ aquaculture or in vitro systems [97,98]. In situ cultivation represents a concrete alternative to the harvesting of biomass from the environment. On the other hand, diseases, adverse climatic events, and predators hamper the in situ, large-scale production. In recent years, successful attempts have been made to set up in vitro culture systems for the cultivation of sponge cells, aiming at the production of sponge metabolites under completely controlled conditions [97]. However, due to the natural monolayer growth of sponge cells, only in few cases have the desired products have been detected. Alternatively, organotypic culture systems, which maintain or mimic the natural tissue structure, have been developed and demonstrate a promising way towards the bioprospecting of sponges [97,99]. To date, no continuously growing sponge cell line has been established. In addition, sponges host a wide diversity of microorganisms that can amount to up to 40–60% of the biomass of the sponge [100]. This complex interaction between sponges and the associated micro-symbionts, as well as the exact origin of active metabolites, remains to be investigated, often because of a general inability to cultivate in axenic conditions the suspected producers. However, in some cases, authors demonstrated that the real metabolite producers are the associated microbes. A clear example was provided by M. Wilson, J. Piel and colleagues which reported the single-cell- and metagenomics-based discovery of the uncultured genus *Entotheonella* associated with the marine sponge *Theonella swinhoei.* The authors detected, in *Entotheonella*, the presence of biosynthetic gene clusters that were responsible for the production of a wide range of cytotoxic compounds (originally isolated from the sponge) [101]. Similarly, the biosynthetic origin of the nucleosides spongosine and other structurally-related compounds that were previously isolated from the sponge *Tethya crypta* was found in a strain of *Vibrio harveyi*, that was associated with *T. crypta* [102]. This enigma has been deeply discussed in numerous works, highlighting the fundamental role of microbes in producing new compounds including clinically approved and preclinically investigated marine natural products, and analyzing the different methods for the cultivation of untapped microorganisms [103,104,105,106,107,108,109]. However, the elucidation of the sponge-microbes relationship is important to decide if it is worth investing in the massive production of sponges or in the isolation and cultivation of the associate microbes. Together with the chemical synthesis of new sponge-derived compounds, the large-scale cultivation of associated microorganisms that produce interesting metabolites, and/or the expression of biosynthetic pathways of interest in easily cultivable hosts, could overcome the issue of the supply problem, filling the gaps that are represented by sustainable sponge cultivation or by the low yield of compounds.

Besides several issues that are related to the bioprospecting of sponges, these organisms still represent a valuable and unexplored source of new potential therapeutic molecules. Further research and developments in the biotechnological production of sponge metabolites can lead to the discovery and production of new effective drugs, the Mediterranean Sea represents a promising reservoir for sponge-derived compounds.

## Figures and Tables

**Figure 1 pharmaceuticals-14-01329-f001:**
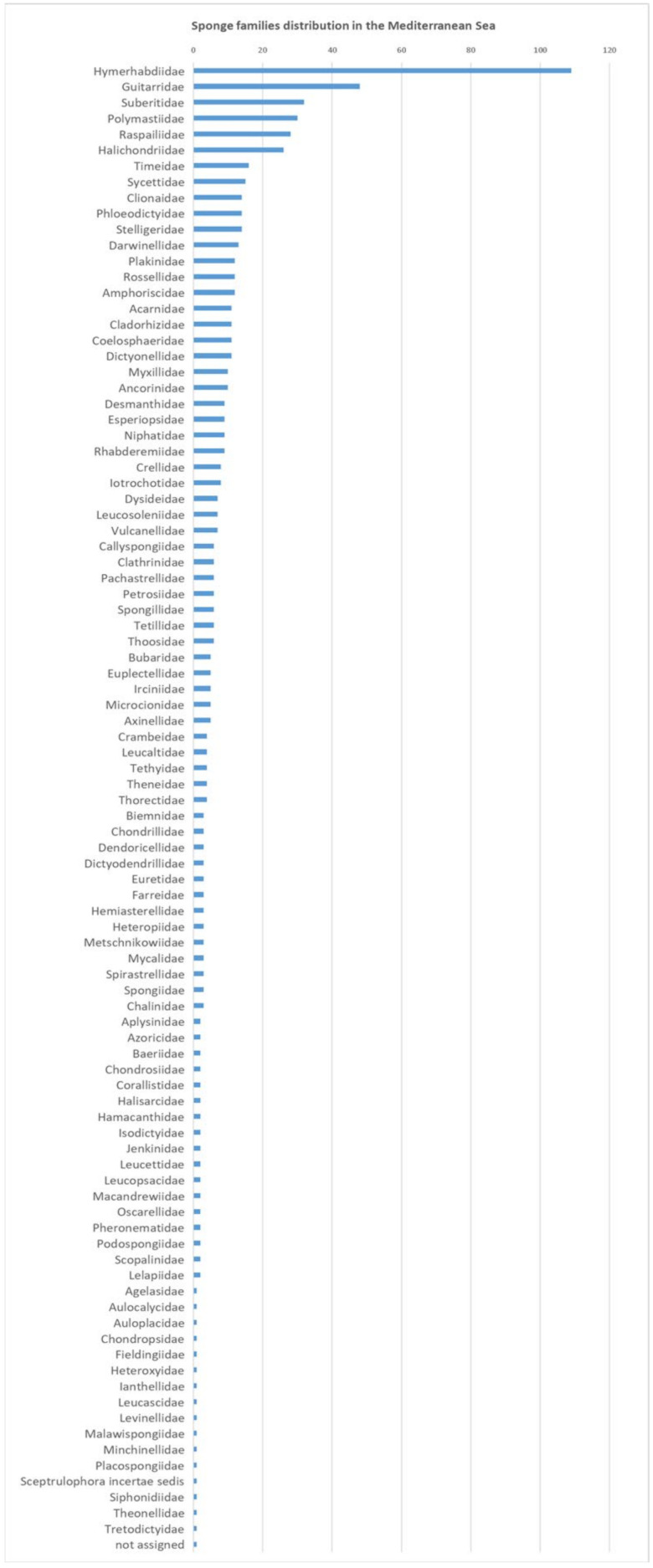
Sponge families that are present in the Mediterranean Sea.

**Figure 2 pharmaceuticals-14-01329-f002:**
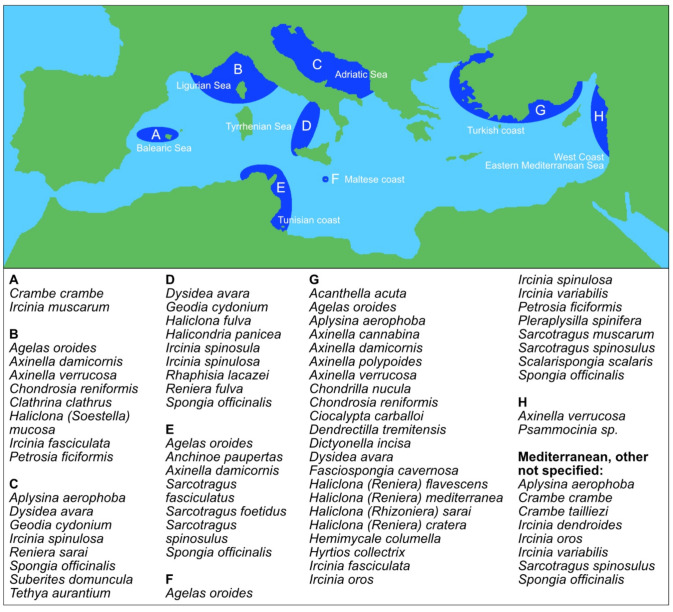
Mediterranean sponges grouped by geographical localization.

**Figure 3 pharmaceuticals-14-01329-f003:**
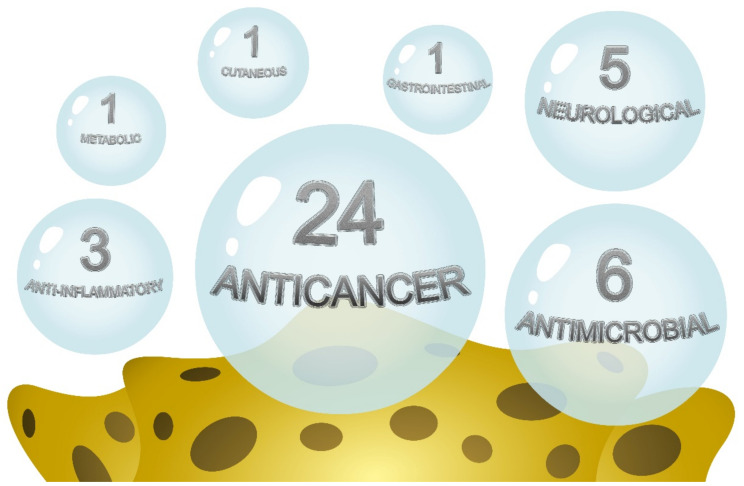
Graphical representation of sponge activity reports for the therapeutic category. The size of the bubble is proportional to the number of papers individuated.

**Figure 4 pharmaceuticals-14-01329-f004:**
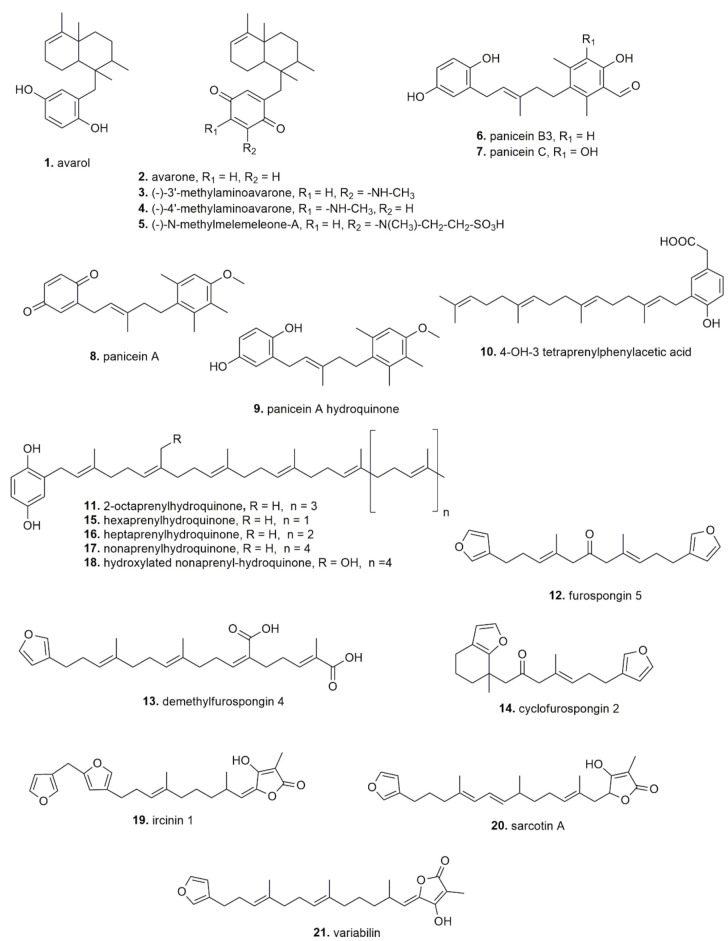
Structures of the molecules that have been isolated from Mediterranean sponges.

**Table 1 pharmaceuticals-14-01329-t001:** Number of accepted aquatic sponge in the world.

Name	Acc. Species	Acc. Species Marine	Acc. Species Fresh
Phylum: Porifera	9436	9174	264
Class: Calcarea	801	801	0
Class: Demospongiae	7817	7555	264
Class: Hexactinellida	687	687	0
Class: Homoscleromorpha	130	130	0

acc. species: the number of accepted species within the specific rank; acc. species marine: number of accepted marine species within the specific rank; acc. species fresh: number of accepted freshwater species within the specific rank.

**Table 2 pharmaceuticals-14-01329-t002:** Health beneficial properties of Mediterranean sponges.

Sponge	Extract/ Compounds (Collection Zone)	In Vitro/ Ex Vivo/ In Vivo	Effect	Active Dose/ Concentration	Reference
*Agelas oroides*	Methanol extract (Portofino’s promontory, Italy)	In vitro: LAN5 and SK-N-BE(2)-C cells	Anticancer	10–20 ppm significantly increased cell death	[24]
	Oroidin, 2-cyano-4,5-dibromopyrrole (Maltese Sea, at Rdum il Bies)	In vitro: KB, Lu1, KB-V, LNCaP, ZR-75-1 cells	Anticancer	Only 2-cyano-4,5-dibromopyrrole: IC_50_ = 1.7–10.8 µg/mL	[25]
	Ethyl acetate extract, Mixture of brominated pyrrole alkaloids (Monastir, Tunisia)	In vitro: *S. epidermidis*, *S. aureus*, *M. luteus*, *E. feacalis*, *E. coli, P. aeruginosa, S. thyphymerium, L. monocytogenes. C. albicans, C. krusei, C. parapsilosis, C. glabrata and C. dubliniensis*	Antimicrobial	7–18 mm inhibition zone (5 mg/disk for bacteria and 10 mg/disk for yeasts)	[26]
*Anchinoe paupertas*	Zarzissine (Zarzis, Tunisia)	In vitro: P-388, KB, NSCLC-N6 cells	Anticancer	IC_50_ = 5–12 µg/mL	[27]
*Aplysina aerophoba*	Aeroplysinin-1; Isofistularin-3 (Adriatic Sea, Kotor Bay, Montenegro)	In vitro: SH-SY5Y, MCF-7 cells	Anticancer	IC_50_ about 5 µM for aeroplysinin-1 on SH-SY5Ycells; IC_50_ > 25 µM for Isofistularin-3 on MCF-7 cells	[28]
	Isofistularin-3 (Mediterranean Sea, unspecified area)	In vitro: RAJI, U-937, JURKAT, K-562, MEG-01, HL-60, SH-SY5Y, PC-3, MDA-MB-231 cells	Anticancer	IC50 = 8.1–50 μM	[29]
	Methanol extract, Aeroplysinin-1 (Rovinj, Croatia)	In vitro: *B. cereus*, *B. subtilis*, *S. aureus*, *S. albus*, *V. anguillarum*, *Flexibacter* sp., *Moraxella* sp.	Antimicrobial	8–30 mm inhibition zone (100 μg/disc)	[30]
*Axinella damicornis*	Damipipecolin and damituricin (Corsica island, Italy)	In vitro: rat neurons	Reversion of the increase of [Ca^2+^] induced by serotonin	IC_50_ = 0.1 µg/mL	[31]
	Ethyl acetate extract, Mixture of brominated pyrrole alkaloids (Monastir, Tunisia)	In vitro, *S. epidermidis*, *S. aureus*, *M. luteus*, *E. feacalis*, *E. coli*, *P. aeruginosa*, *S*. *thyphymerium*, *L. monocytogenes*, *C. tropicalis*	Antimicrobial	7–26 mm inhibition zone (5 mg/disk)	[26]
*Axinella verrucosa*	Alkaloids (Corsica island, Italy)	In vitro: rat neurons	Reversion of the increase of [Ca^2+^] induced by serotonin, glutamic and quisqualic acid	IC_50_ = 10 µg/mL	[32]
	Methanol extract, Hymenialdisine, 10- E-Hymenialdisine, Spongiacidine B (Latakia coast, Syria)	In vitro, *S. aureus*, *A. Septicus*, *P. vulgaris*, *P. aeruginosa*	Antimicrobial	>20 mm of inhibition zone (concentration not determined)	[33]
*Chondrosia reniformis*	Marine collagen hydrolysates (Peptides rich in hydroxyproline)(Portofino, Italy)	In vitro: L929 and HaCaT cells	Enhanced proliferation of fibroblasts and keratinocytes; Enhanced collagen 1A expression and release;Protected from UV damages (cell death, keratin 1 and 10 expression);Promoted wound healing	IC_50_ = 10, 50, or 100 µg/mL	[34]
	Proteic P4 fraction (from a crude extract) containing Chondrosin (Portofino, Italy)	In vitro: L929, RAW 264.7, MDA-MB-468 and HeLa cells	Anticancer	1–100 µg/mL	[35]
*Clathrina clathrus*	CH2Cl2/MeOH extract, Clathridimine, clathridine, clathridine zinc complex, preclathridine (Marseille, France)	In vitro, *S. aureus*, *E. coli*, *C. albicans*	Antimicrobial	11–38 mm inhibition zone (concentration not determined)	[36]
*Crambe crambe*	Crambescidin 816 (Favignana island, Italy)	In vitro/ex vivo, (1) HCT-16 cells; (2) NG 108-15 cells; (3) Guinea pig ileum	(1) Cytotoxicity against colon cells; (2) potent Ca^2+^ antagonist activity; (3) inhibition of acetylcholine-induced contraction of ileum	(1) IC_50_ = 0.24 ug/mL; (2) EC_50_ = 1.5 × 10^−4^ µM; (3) −30% at 6 pM	[37]
	Crambescidin 816, 844 and 800 (Isla de Formentor, Cueva, Spain)	In vitro: L-1210 cells	Anticancer	98% cytotoxicity at 0.1 µg/mL	[38]
	Crambescidin 816, 830 and 800 (Mediterranean Sea, unspecified area)	In vitro: HepG2 In vivo: zebrafish xenografted colon cancer	Anticancer	IC_50_ = 0.18–2.66 μM Significant tumor growth decrease at 1 and 2 mM of embryos treatment	[39]
*Crambe tailliezi*	P3 compound (Mediterranean Sea, unspecified area)	In vitro: U-2 OS cells	Anticancer	IC_50_ = 6.6 μM	[40]
*Dysidea avara*	Avarol (Mediterranean Sea, unspecified area)	In vivo, mouse	Reduction of paw edema	ED_50_: 9.2 mg/kg (orally) and 97 μg/ear (topically)	[41]
	Avarol (Mediterranean Sea, unspecified area)	In vitro, human recombinant enzyme	Inhibition of human recombinant synovial PLA2 activity	IC_50_: 158 μM	
	Avarone (Mediterranean Sea, unspecified area)	In vivo, mouse	Reduction of ear edema	ED_50_: 4.6 mg/kg (orally) and 397 µg/ear (topically)	
	Avarol and avarone (Mediterranean Sea, unspecified area)	In vitro, human leukocytes	Inhibition of LTB4 and TXB2 release	IC_50_: 0.6 and 0.8 μM (LTB4); IC_50_ 1.4 and 3.3 μM TXB2)	
	Etanolic extract (Dardanelles, Turkey)	In vitro: K562, KMS-12PE, A549, A375, H929, MCF7, HeLa, HCT116 cells	Anticancer Inhibition tyrosine kinase	IC_50_ = 2.91–25.15 µg/mL; Inhibition PDGFRβ (2.57 μg/mL)	[42]
	Avarol (Bay of Naples, Italy)	HT-29 cells	Anticancer	IC_50_ = <7 μM	[43]
	Avarol; Avarone (Bay of Kotor, Montenegro)	In vitro: L5178y cells In vivo: mouse	Anticancer	IC_50_ = 0.93 μM (avarol) and 0.62 μM (avarone) 10 mg/kg	[44]
	Avarol; Avarone (Bay of Naples, Italy)	In vitro: L1210, Raji C8166 cells	Anticancer	IC_50_ = 9.2–18.1 μM	[45]
	Avarol; Avarone; (−)-3′-methylaminoavarone; (−)-4′-methylaminoavarone; N-methylmelemeleone-A; *(Fethiye, Turkey)*	In vitro: HCT116 H4IIE cells	Anticancer	Avarone most potent: IC_50_ = 5.3–5.5 μM	[46]
	Avarol (Bay of Naples, Italy)	In vitro: *C. albicans MH2, C. albicans 4/07*, *C. albicans 4/16*, *C. albicans 2/14*, *C. glabrata*, *C. krusei*, *C. albicans ATCC 10231*, *C. tropicalis ATCC 750*	Antimicrobial	MIC and MFC: 0.8–12 µg/mL	[47]
*Fasciospongia* *cavernosa*	Carvenolide (Aegean Sea)	In vitro, human recombinant enzyme	Inhibition of human synovial PLA2	IC_50_: 8.8 μM	[48]
		In vitro, human macrophages	Reduction of TNF-α, nitrite and PGE_2_ production	4.9 μM (TNFα); 7.7 μM (nitrite); IC_50_: 9.3 μM (PGE2)	
*Geodia cydonium*	MeOH fraction (Bay of Naples, Italy)	In vitro: MCF-7, MDA-MB231, and MDA-MB468	Anticancer	IC_50_ = 44–80 µg/mL	[49]
*Haliclona fulva*	Butanolic extract, Fulvynes A-I (linear polyoxygenated acetylene) (Bay of Naples, Italy)	In vitro, chloramphenicol-resistant *Bacillus subtilis*	Antimicrobial	IC_50_: 60–12 µM	[50]
*Haliclona (Soestella) mucosa*	Panicein A hydroquinone (Villefranche-sur-Mer, France)	In vitro: MeWo cells	Anticancer	Panicein A: IC_50_ = >30 µM	[51]
*Ircinia dendroides*	Palinurin (Mediterranean Sea, unspecified area)	In vitro, neuroblastoma SH-SY5Y cells	rGSK-3b inhibitory activity	IC_50_ = 2.6 µM	[52]
*Ircinia fasciculata*	Polyprenyl-1,4-hydroquinone derivates (hexa-, hepta- and nona- prenyl-1,4-hydroquinone) (Fethiye, Turkey)	In vitro: H4IIE cells	Anticancer	Heptaprenyl hydroquinone: IC_50_ = 2.5 μM	[53]
*Ircinia muscarum*	Polyprenyl-hydroquinones; furanoterpenoids (Baleares, Spain)	In vitro: Inhibition of CDC25 phosphatase	Anticancer	4-OH-3 Tetraprenylphenylacetic acid: IC_50_ = 0.4–4 µM; 2-Octaprenylhydroquinone: IC_50_ = 400 µM	[54]
*Ircinia spinulosa*	Polyprenyl-hydroquinones; furanoterpenoids(Bodrum, Turkey; Naples, Italy; Sutomiscica, Croatia)	In vitro: Inhibition of CDC25 phosphatase	Anticancer	4-OH-3 Tetraprenylphenylacetic acid: IC_50_ = 0.4–4 µM;2-Octaprenylhydroquinone: IC_50_ = 400 µM	[54]
*lrcinia spinosula*	IS2, IS3 (Bay of Naples, Italy)	In vitro: human recombinant enzyme	Inhibition of synovial PLA2	IC_50_: 48.7 and 48 µM	[55]
		In vitro: human neutrophils	Inhibition of LTB4 production and TXB2 synthesis and release	IC_50_: 23.1 and 7.4 µM (LTB4); IC_50_ 3.9 and 3.4 µM (TXB2)	
		In vivo, mouse	Reduction of ear inflammation	250 µg/ear and 125 µg/ear (topically)	
*Petrosia ficiformis sp.*	Methanol extract (Portofino’s promontory, Italy)	In vitro: LAN5 and SK-N-BE(2)-C cells	Anticancer	10–20 ppm significantly increased cell death	[24]
*Rhaphisia iacazei*	Topsentin B1 and B2 (Ustica, Italy)	In vitro: NSCLC-N6 cells	Anticancer	IC_50_ = 6.3 (B1) and 12 (B2) µg/mL	[56]
*Reniera fulva*	Paniceins A, Panicein B3, Panicein C (Favignana island, Italy)	In vitro: CCRF-CEM, NCI-H522	Anticancer	−log_10_ I_50_: 5.11–5.48	[57]
*Reniera sarai*	3-Alkylpyridinium polymers (Northern Adriatic Sea)	In vitro, isolated enzyme	AChE inhibition	50% inhibition induced by 0.06 (human erythrocyte AChE), 0.08 (electric eel AChE), 0.7 (insect recombinant AChE) and 0.14 µg/mL (horse serum butyrylcholinesterase)	[58]
	3-Alkylpyridinium oligomers and polymers (Northern Adriatic Sea)	Ex vivo, mouse skeletal muscle	AChE inhibition; blockade of the neuromuscular transmission	IC_50_ = 18.5 μM (mouse muscle twitch), 18.5 μM (tetanic contraction)	[59]
*Sarcotragus* *fasciculatus*	Furanosesterterpene tetronic acids and Polyprenyl-hydroquinones (Monastir, Tunisia)	In vitro: A549, Hs683, MCF-7, SKMEL-28, U373, B16F10)	Anticancer	Furanosesterpene tetronic acids IC_50_=> 80 µM; Polyprenyl-hydroquinones IC_50_ = 3–23 µM	[60]
*Sarcotragus* *foetidus*	Furanosesterterpene tetronic acids and Polyprenyl-hydroquinones (Cap Zebib, Tunisia)	In vitro: A549, Hs683, MCF-7, SKMEL-28, U373, B16F10)	Anticancer	Furanosesterpene tetronic acids IC_50_ => 80 µM; Polyprenyl-hydroquinones IC_50_ = 3–23 µM	[60]
*Sarcotragus muscarum*	Polyprenyl-1,4-hydroquinone derivates (hexa-, hepta- and nona- prenyl-1,4-hydroquinone)(Mersin, Turkey)	In vitro: H4IIE cells	Anticancer	Heptaprenylhydroquinone: IC_50_ = 2.5 μM	[53]
*Sarcotragus spinosulus*	Hydroxylated nonaprenylhydroquinone; hepta- and octa-prenylhydroquinone (Callejones, Spain)	In vitro: K562 cells	Anticancer (apoptosis)	Hepta- and octa-prenylhydroquinone: 8 and 10 µM; Hydroxylated nonaprenylhydroquinone 193 µM	[61]
	Furanosesterterpene tetronic acids and Polyprenyl-hydroquinones (Tabarka and Monastir, Tunisia)	In vitro: A549, Hs683, MCF-7, SKMEL-28, U373, B16F10)	Anticancer (apoptosis)	Furanosesterpene tetronic acids IC_50_ => 80 µM; Polyprenyl-hydroquinones IC_50_ = 3–23 µM	[60]
*Spongia officinalis*	Methanol and fractions (0%, 50% and 80% MeOH in water)(Monastir, Tunisia)	In vitro: A549, HCT15, MCF7 cells	Anticancer	F3 fraction greater potency: IC_50_ = 212–572 µg/mL	[62]
	Furospongin-5; Cyclofurospongin-2; demethylfurospongin-4 (Cadiz, Spain)	In vitro: P-388 cells	Anticancer	Furospongin: IC_50_ = 5 µg/mL	[63]
	Polyprenyl-hydroquinones; furanoterpenoids (Bodrum, Turkey)	In vitro: Inhibition of CDC25 phosphatase	Anticancer	4-OH-3 Tetraprenylphenylacetic acid: IC_50_ = 0.4–4 µM; 2-Octaprenylhydroquinone: IC_50_ = 400 µM	[54]
	Methanol/water, crude extract and fractions (Tunisia)	In vivo, mice	Anticovulsant and analgesic	600 mg/kg crude extract; 100 and 200 mg/kg 50% methanol fractions	[64]

## Data Availability

No new data were created or analyzed in this study. Data sharing is not applicable to this article.

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
