# Peer review of "Pharmacological Activities of Extracts and Compounds Isolated from Mediterranean Sponge Sources"

_pharmaceuticals, 2021, doi:10.3390/ph14121329_

Round 1
Reviewer 1 Report
Please see following report and suggestions s to how minor changes would enable the paper to be published. Since it cannot be published as is, I have used a lower score. With some very simple changes, it is definitely publishable.
Although this is a nice (and up to date in some cases) compendium of the pharmacological activities reported from Mediterranean sponges, be they single isolated and purified agents or in other cases, particularly in the case of antimicrobial activities, crude extracts, to my mind there is a potential fatal flaw in the compendium from a pharmacological aspect.
That flaw, though mentioned “en passant” in the text is that as far as I am aware, to date, no isolated axenic sponge cell line has produced a verifiable single pharmacologically active agent, though a fair number of sponge experts have tried to do so, none have yet reported a defined success.
Since sponges, irrespective of Genus/Species contain around 50% by weight of single-celled organisms, mainly microbes and fungi, though some may also have protists as well, although the authors mention the possibilities of some single cell production, they do not appear to take into account the work of the Piel group in Switzerland (using Palauan and Japanese sponges) reported in 2013/4 where similar genera, and with many subsequent reports demonstrating similar activities in sponges in most of the world’s oceans/seas) demonstrating that the main producer of the cytotoxic agents isolated from the Palauan/Japanese sponges is an as yet uncultivated single celled microbe that produces the majority of the purified cytotoxins reported from that genus.
Similar results have been seen with other agents/sponges and reported in the last 6 plus years. In addition, the same basic “sponge microbe” reported by the Piel group produces a series of compounds, the onnamides, where the general structure is effectively that of the mycalamides from a cold-water Mycale species in New Zealand, and even more “surprisingly” from a terrestrial pseudomonad that produces the Amazonian beetle toxin, Pederine.
In addition, the arabinosides reported by Bergmann in the middle 1950s are almost certainly produced by a Vibrio species reported by the Gerwick lab in 2015 isolated from the same sponge species/site as the original Bergmann collection almost 60 years previously.
Thus, if the authors want to alter the title of their paper to: “Pharmacological Activities of Extracts and Compounds Isolated from Mediterranean Sponge Sources” then the article would be well received. To make the unwarranted assumption that the molecules are produced by the sponge itself and not by either the microbes within it acting singularly or in concert (since interplay amongst microbes is well known) is not in any way justified in light of the current state of knowledge.
In addition, I would recommend that when reporting antimicrobial activity where the crude extract is measured in tens of mgs/ml, the material is effectively inactive.
Author Response
#Referee 1
Please see following report and suggestions s to how minor changes would enable the paper to be published. Since it cannot be published as is, I have used a lower score. With some very simple changes, it is definitely publishable.
Although this is a nice (and up to date in some cases) compendium of the pharmacological activities reported from Mediterranean sponges, be they single isolated and purified agents or in other cases, particularly in the case of antimicrobial activities, crude extracts, to my mind there is a potential fatal flaw in the compendium from a pharmacological aspect. That flaw, though mentioned “en passant” in the text is that as far as I am aware, to date, no isolated axenic sponge cell line has produced a verifiable single pharmacologically active agent, though a fair number of sponge experts have tried to do so, none have yet reported a defined success. Since sponges, irrespective of Genus/Species contain around 50% by weight of single-celled organisms, mainly microbes and fungi, though some may also have protists as well, although the authors mention the possibilities of some single cell production, they do not appear to take into account the work of the Piel group in Switzerland (using Palauan and Japanese sponges) reported in 2013/4 where similar genera, and with many subsequent reports demonstrating similar activities in sponges in most of the world’s oceans/seas) demonstrating that the main producer of the cytotoxic agents isolated from the Palauan/Japanese sponges is an as yet uncultivated single celled microbe that produces the majority of the purified cytotoxins reported from that genus. Similar results have been seen with other agents/sponges and reported in the last 6 plus years. In addition, the same basic “sponge microbe” reported by the Piel group produces a series of compounds, the onnamides, where the general structure is effectively that of the mycalamides from a cold-water Mycale species in New Zealand, and even more “surprisingly” from a terrestrial pseudomonad that produces the Amazonian beetle toxin, Pederine. In addition, the arabinosides reported by Bergmann in the middle 1950s are almost certainly produced by a Vibrio species reported by the Gerwick lab in 2015 isolated from the same sponge species/site as the original Bergmann collection almost 60 years previously.
Response: We thank the reviewer for this precise observation. It is true that the argument of associated microbes and the true producers of compounds has not been discussed as much as it deserves. The reason is that there is so much literature on this topic that it would deserve an additional review and it would move our focus, which is bring more attention on the Mediterranean sponges and its field of research. However, we think that the comment of this reviewer is right and so we implemented and revised part of the conclusion and perspectives (starting from about line 600), adding some relevant references. We hope that this integration is fine.
Thus, if the authors want to alter the title of their paper to: “Pharmacological Activities of Extracts and Compounds Isolated from Mediterranean Sponge Sources” then the article would be well received. To make the unwarranted assumption that the molecules are produced by the sponge itself and not by either the microbes within it acting singularly or in concert (since interplay amongst microbes is well known) is not in any way justified in light of the current state of knowledge.
Response: The title was changed following the reviewer suggestion.
In addition, I would recommend that when reporting antimicrobial activity where the crude extract is measured in tens of mgs/ml, the material is effectively inactive.
Response: We agree with this comment. We reported activity in µg/mL or mg/disk.
Thank you for the kind consideration of our manuscript.
Sincerely,
Lorenzo Di Cesare Mannelli
Reviewer 2 Report
I commend the authors for writing a comprehensive review manuscript on the natural products of sponges from Mediterranean Sea and their pharmacology. It was well-written and I enjoyed reading the manuscript. It highlighted the importance of chemistry and the pharmacology of sponge natural products towards drug discovery and development.
Here are the strengths of the manuscript.
- It has a well-defined topic and issue.
- The organization of the manuscript (topic and sub-topic) is logical. The structure and flow are outstanding.
- There is depth and accuracy of the discussion on the critical issues.
- The review identified gaps and the best avenues for future research.
One weakness of the manuscript is the presentation of data. Below are my specific comments and suggestions.
- Figure 1 is hard to follow. Too many shades of blue, yellow, orange, and green etc. Please revise the graph and annotate it well.
- Since the readership of MDPI is broad, some readers might not be knowledgeable of the geography in the Mediterranean Sea. Indicate the name for each site instead of plainly annotating with letters and shading in the map in Figure 2. For example, A-Balearic Sea, B-Ligurian Sea, and others.
- Figure 3 - Indicate the total number of papers that reported therapeutic activity and indicate the number of papers per bubble.
- Table 2 - insert column for collection site in Mediterranean Sea.
line 519 - IC50 2.6 M??? It appears that µ is missing next to M.
table 2 - page 14 - top most row. Check units for concentration. It appears that µ is missing next to M.
table 2 - page 15 Dysidea avara row. Check units for concentration. It appears that µ is missing next to M or g.
table 2 - page 16 rows 5 to 7, and rows 10. Check units for concentration. It appears that µ is missing next to M or g.
Author Response
#Referee 2
I commend the authors for writing a comprehensive review manuscript on the natural products of sponges from Mediterranean Sea and their pharmacology. It was well-written and I enjoyed reading the manuscript. It highlighted the importance of chemistry and the pharmacology of sponge natural products towards drug discovery and development. Here are the strengths of the manuscript.
- It has a well-defined topic and issue.
- The organization of the manuscript (topic and sub-topic) is logical. The structure and flow are outstanding.
- There is depth and accuracy of the discussion on the critical issues.
- The review identified gaps and the best avenues for future research.
One weakness of the manuscript is the presentation of data. Below are my specific comments and suggestions.
- Figure 1 is hard to follow. Too many shades of blue, yellow, orange, and green etc. Please revise the graph and annotate it well.
Response: As suggested Figure 1 was completely revised with a different graphical approach.
- Since the readership of MDPI is broad, some readers might not be knowledgeable of the geography in the Mediterranean Sea. Indicate the name for each site instead of plainly annotating with letters and shading in the map in Figure 2. For example, A-Balearic Sea, B-Ligurian Sea, and others.
- Figure 3 - Indicate the total number of papers that reported therapeutic activity and indicate the number of papers per bubble.
- Table 2 - insert column for collection site in Mediterranean Sea.
line 519 - IC50 2.6 M??? It appears that µ is missing next to M.
table 2 - page 14 - top most row. Check units for concentration. It appears that µ is missing next to M.
table 2 - page 15 Dysidea avara row. Check units for concentration. It appears that µ is missing next to M or g.
table 2 - page 16 rows 5 to 7, and rows 10. Check units for concentration. It appears that µ is missing next to M or g.
Response: All these points were corrected.
Thank you for the kind consideration of our manuscript.
Sincerely,
Lorenzo Di Cesare Mannelli
Reviewer 3 Report
Dear Editor
- I wonder, why not summarize some skeleton together, I mean to draw the similar skeletons to draw it in one skeleton.
I.e. Compounds 11, 16, 17 and 18 to put them in one skeleton and so on.
- May be there are some compounds can be used as a marker for taxonomy or pharmaceutical different effect. Would you mind add it to the conclusion part.
There are some typing mistakes we discuss as fellow:
Line 2 Natural Products from Marine Sponges and Their Pharmacological Applications: A Focus on The Mediterranean Sea
Line 51 Because sponges cannot move and lack physical
Because of sponges cannot move and lack of physical
Line 54 offspring
off spring
Line 165 From Dysidea avara collected in the bay of Kotor, ex Ju- goslavia, two secondary sesquiterpenoid metabolites, that were avarol (1) and its derivative avarone (2) (previously chemically-characterized by Minale et al., 1974 [28]) were isolated by Müller and collaborators [28].
Line 236 In vitro experi-
In vitro experi-
Line 246 4,S-guanidino-pyridazine compound
4S-guanidino-pyridazine compound
Line 249 Rhaphisia lacazei collected in the
Rhaphisia lacazei was collected in the
Line 302 To counter this phenomenon, the hunt for
To count this phenomenon,
Line 360 From the butanolic extract of this spe-
From The butanolic extract of this spe-
Line 404 ish marine environment [63], [79].
Line 435 none (11) (IS2) and 2-[24-hydroxy]-octaprenyl-1,4-hydroquinone (50) (IS3), isolated from
Line 435 none (11) (IS2) and 2-[24-hydroxy]-octaprenyl-1,4-hydroquinone (50) (IS3), isolated from
Line 477 Da molecular (what does it mean).
Line 488 were reported in rat ex vivo evaluation using poly-APS higher than
Finally, the review document supports the future working in the field of marine sponge in the Mediterranean Sea with fully hope to accept my comments. Overall the review can be accepted after the minor revision.
Author Response
#Referee 3
- I wonder, why not summarize some skeleton together, I mean to draw the similar skeletons to draw it in one skeleton.
I.e. Compounds 11, 16, 17 and 18 to put them in one skeleton and so on.
Response: Thank you for this comment. When possible the structures and their analogues have been put in one skeleton, reducing also the number of the figures, from 4 to 3.
May be there are some compounds can be used as a marker for taxonomy or pharmaceutical different effect. Would you mind add it to the conclusion part.
Response: Thank you for this indication. We mentioned in the conclusion Avarol, its derivatives, aeroplysinin, and the class of alkaloids in general as the most active molecules
There are some typing mistakes we discuss as fellow:
Line 2 Natural Products from Marine Sponges and Their Pharmacological Applications: A Focus on The Mediterranean Sea
Response: The title has been changed and revised
Line 51 Because sponges cannot move and lack physical
Because of sponges cannot move and lack of physical
Line 54 offspring
off spring
Line 165 From Dysidea avara collected in the bay of Kotor, ex Ju- goslavia, two secondary sesquiterpenoid metabolites, that were avarol (1) and its derivative avarone (2) (previously chemically-characterized by Minale et al., 1974 [28]) were isolated by Müller and collaborators [28].
Line 236 In vitro experi-
In vitro experi-
Line 246 4,S-guanidino-pyridazine compound
4S-guanidino-pyridazine compound
Line 249 Rhaphisia lacazei collected in the
Rhaphisia lacazei was collected in the
Response: The text has been revised and corrected
Line 302 To counter this phenomenon, the hunt for To count this phenomenon,
Response: The meaning of this sentence was “to fight this phenomenon (the AMR)”, so we corrected in “ to counteract this phenomenon”
Line 360 From the butanolic extract of this spe-
From The butanolic extract of this spe-
Response: The sentence has been corrected and slightly rephrased.
Line 404 ish marine environment [63], [79].
Line 435 none (11) (IS2) and 2-[24-hydroxy]-octaprenyl-1,4-hydroquinone (50) (IS3), isolated from
Line 435 none (11) (IS2) and 2-[24-hydroxy]-octaprenyl-1,4-hydroquinone (50) (IS3), isolated from
Line 477 Da molecular (what does it mean).
Line 488 were reported in rat ex vivo evaluation using poly-APS higher than
Response: The text has been revised and corrected
Finally, the review document supports the future working in the field of marine sponge in the Mediterranean Sea with fully hope to accept my comments. Overall the review can be accepted after the minor revision.
Response: We are grateful to the referee for this positive comment.
Thank you for the kind consideration of our manuscript.
Sincerely,
Lorenzo Di Cesare Mannelli